# Soil Classification from Piezocone Penetration Test Using Fuzzy Clustering and Neuro-Fuzzy Theory

**Joon-Shik Moon [1], Chan-Hong Kim [2] and Young-Sang Kim [3],***

1 Department of Civil Engineering, Kyungpook National University, Daegu 41566, Korea; orangedreamer@gmail.com
2 Korea Mine Rehabilitation and Mineral Resources Corporation, Wonju 26464, Korea; chahkim@komir.or.kr
3 Department of Civil Engineering, Chonnam National University, Gwangju 61186, Korea
* Correspondence: geoyskim@jnu.ac.kr; Tel.: +82-62-530-1654

**Abstract:** The advantage of the piezocone penetration test is a guarantee of continuous data, which are a source of reliable interpretation of the target soil layer. Much research has been carried out for several decades, and several classification charts have been developed to classify in situ soil from the cone penetration test result. Even though most present classification charts or methods were developed on the basis of data which were compiled over many countries, they should be verified to be feasible for local country. However, unfortunately, revision of those charts is quite difficult or almost impossible even though a chart provides misclassified soil class. In this research, a new method for developing soil classification model is proposed by using soft computing theory—fuzzy C-mean clustering and neuro-fuzzy theory—as a function of 5173 piezocone penetration test (PCPT) results and soil boring logs compiled from 17 local sites around Korea. Feasibility of the proposed soil classification model was verified from the viewpoint of accuracy of the classification result by comparing the classification results not only for data which were used for developing the model but also new data, which were not included in developing the model with real boring logs, other fuzzy computing classification models, and Robertson's charts. The biggest advantage of the proposed method is that it is easy to make the piezocone soil classification system more accurate by updating new data.

**Keywords:** piezocone; soil classification; fuzzy C-means clustering; neuro-fuzzy

## 1. Introduction

Underground information is a main factor to be considered during the construction and design phases. In particular, stratigraphy is essential for economical design of a foundation because most construction projects are carried out at the deposit layer on bedrock. Boring logs from subsurface exploration at a constant interval along the project area are the only source of data. They are drawn from some resources such as the penetration rate, soil color, and driller's experience, which is dependent upon their career and, thus, may could not always reflect the nature of the ground. Therefore, penetration tests, such as the cone penetration test (CPT), piezocone penetration test (PCPT), and standard penetration test (SPT), have been used together. PCPT has an advantage in the view of continuity and standardization, even when evaluating interbedded thin layers from thick deposit layers. Research on soil classification from CPT results was commenced by Begemann [1]. Furthermore, Douglas and Olsen [2] developed a new soil classification chart using electric cone penetration test results. After introducing the piezocone which can measure pore pressure readings, many researchers including Robertson et al. [3], Robertson [4], and Jefferies and Davis [5] developed various types of soil classification charts and/or techniques. However, most classification charts provide only soil behavior type, while local engineers who are familiar with Unified Soil Classification System (USCS) have trouble with understanding

relevant results. In addition, the adopted charts and methods sometimes give different soil classification results for the same input parameters, and even two charts developed by one researcher may lead to different soil types. To complement the weakness of the chart type soil classification method, and considering the fuzziness of the ground, there has been progress in studies on soft computing. Pradhan [6] developed fuzzy membership functions on the basis of Robertson et al.'s chart [3], and Zhang and Tumay [7] suggested a fuzzy soil classification method according to Douglas and Olsen's chart [2]. On the other hand, Hegazy and Mayne [8] introduced a clustering method as a function of normalized cone resistance, $Q_t$, and pore pressure ratio, $B_q$. Clustering methods can give soil classifications between upper and lower soil data but not the soil type of each soil datum. On the other hand, soil classification by fuzzy theory can provide soil type to each soil datum and has the advantage of simply being updated for newly acquired soil data. However, it also has a problem that the classification result is highly dependent on the fuzzy membership function. As described before, Pradhan [6] and Zhang and Tumay [7] developed fuzzy membership functions on the basis of charts such as those proposed by Robertson et al., and Douglas and Olson, respectively. Therefore, these fuzzy classifications seemingly remain unable to reflect local soil type. Recently, machine learning has been used to classify soils from CPT data [9–13] and to successfully estimate soil and design parameters [9,13]. Rauter and Tschuchnigg [14] suggested a machine learning classifier based on a support vector machine, artificial neural network, and random forest to predict soil classes according to Oberhollenzer et al. [15] and soil behavior types according to Robertson [16–18]. They showed that machine learning algorithms can classify soils on the basis of grain size distribution and the updated soil behavior classification from Robertson (i.e., SBT, SBTn, ModSBTn). However, since they used cone tip resistance $q_c$, sleeve friction $f_s$, total vertical stress $\sigma_v$, and static pore pressure $u_o$ as input variables, their model can still be improved by adopting pore pressure parameters such as $B_q$.

In this study, a new soil classification method was developed using the neuro-fuzzy technique, in which the membership function was developed by a neural network and not adjusted by the trial-and-error method to present classification charts. Moreover, input variables and relevant soil types were determined on the basis of proximity between compiled soil data using the fuzzy C-mean clustering (FCM) method and not by the developer's experience. To show the feasibility of the proposed model, new PCPT results which were not included in the soil database were classified using the proposed neuro-fuzzy model and compared with Robertson et al.'s chart classification, Pradhan's fuzzy classification, Zhang and Tumay's fuzzy classification, and the Unified Soil Classification System (USCS).

## 2. Soil Classification Method for CPT and PCPT

### 2.1. Soil Classification Charts

Figure 1 shows Robertson et al. [3]'s classification charts as a function of $q_t$, $R_f$, and $B_q$. Their definitions are as follows:

$$q_t = q_c + (1 - a)u_{bt}, \tag{1}$$

$$R_f = \frac{f_s}{q_c} \times 100(\%), \tag{2}$$

$$B_q = \frac{(u_{bt} - u_o)}{(q_t - \sigma_{vo})}, \tag{3}$$

where $q_t$ is the corrected cone tip resistance, $q_c$ is the measured cone tip resistance, $u_{bt}$ is the penetration-induced pore pressure measured behind the cone tip, a is the unequal area ratio, $R_f$ is the friction ratio, $f_s$ is the sleeve friction, $B_q$ is the pore pressure ratio, $u_o$ is the static pore pressure before cone penetration, and $\sigma_{vo}$ is the total stress. Robertson's charts have been widely used, and their feasibility was verified by several researchers.

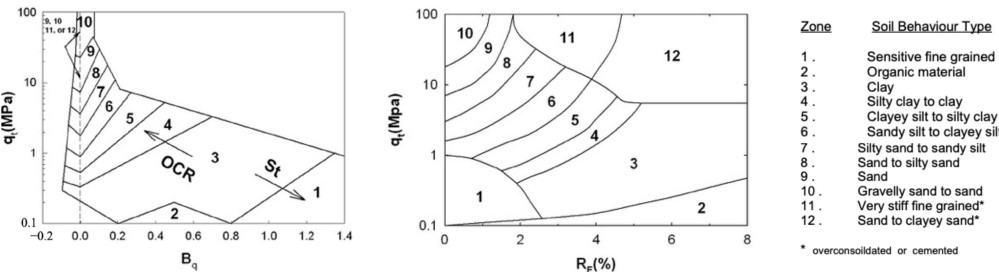

**Figure 1.** Soil classification charts by Robertson et al. [3].

## 2.2. Soil Classification Using Fuzzy Theory

Natural phenomena are known not to be decided in absolute terms such as 0 or 1. Zadeh [19] introduced "soft computing", the concept of fuzzy theory to describe nature's ambiguousness. This theory can present intermediate values using the fuzzy membership function. Various soft computing methods have been suggested after Zadeh [19], and studies on soil classification from CPT and PCPT using soft computing are summarized below.

### 2.2.1. Pradhan's Study

Pradhan [6] suggested a soil classification method using fuzzy theory. He developed fuzzy membership functions for input variables $q_t$, $F_r(= f_s/q_t)$, and $B_q$ according to Robertson et al. [2]. However, soil types were only classified into "clay", "silt", and "sand". The maximum grade for membership functions was limited to 0.8 when considering uncertainty in soil classification. Figure 2 shows the membership functions of three soil sets in terms of $q_t$, $B_q$, and $F_r$ respectively. The reader is referred to Pradhan [6] for the detailed expression of fuzzy membership functions.

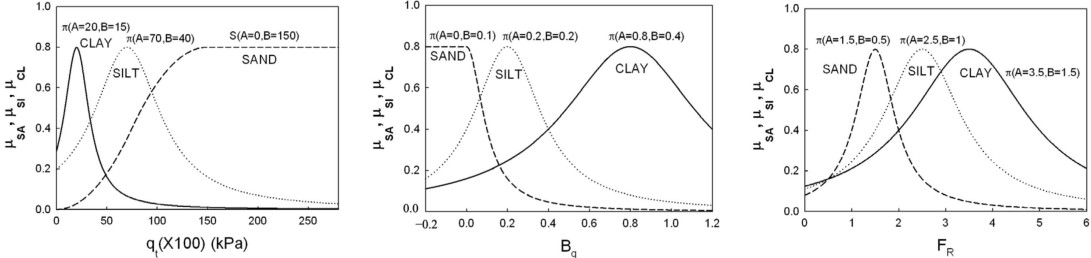

**Figure 2.** Fuzzy membership functions of soil sets for three parameters, $q_t$, $F_r$, and $B_q$.

Three soil fuzzy sets for clay, silt, and sand were named CL, SI, and SA, as defined in Equation (4). $\mu_{CL}$, $\mu_{SI}$, and $\mu_{SA}$ represent the summation of membership function values from each chart for clay, silt, and sand, respectively.

$$\begin{aligned}
\text{Clayey soil}: \ \text{CL} &= \sum \mu_{CL}(a_i)/a_i (i = 1, 2, 3), \\
\text{Silty soil}: \ \text{SI} &= \sum \mu_{SI}(a_i)/a_i (i = 1, 2, 3), \\
\text{Sandy soil}: \ \text{SA} &= \sum \mu_{SA}(a_i)/a_i (i = 1, 2, 3),
\end{aligned} \tag{4}$$

where $a_1 = q_t$, $a_2 = F_r$, and $a_3 = B_q$.

### 2.2.2. Zhang and Tumay's Study

Zhang and Tumay [7] grouped soil into three types, i.e., HPC (highly probable clay), HPM (highly probable mixed soil), and HPS (highly probable sand) on the basis of the Unified Soil Classification System (USCS) and used $q_c$ and $R_f$ as input variables. They suggested new fuzzy membership functions $\mu_C(U)$, $\mu_m(U)$, and $\mu_s(U)$ as shown in Figure 3, with an intermediate soil classification index ($U$) empirically based on Douglas and Olsen [2]'s chart.

$$U = \frac{(a_1 X - a_2 Y + b_1)(c_1 X - c_2 Y + d_1)}{(c_1 X - c_2 Y + d_1)^2 + (c_2 X + c_1 Y + d_2)^2} - \frac{(a_2 X + a_1 Y + b_2)(c_2 X + c_1 Y + d_2)}{(c_1 X - c_2 Y + d_1)^2 + (c_2 X + c_1 Y + d_2)^2},\tag{5}$$

where $X = 0.1539 R_f + 0.8870 \log q_c - 3.35$, and $Y = -0.2957 R_f + 0.4617 \log q_c - 0.37$.

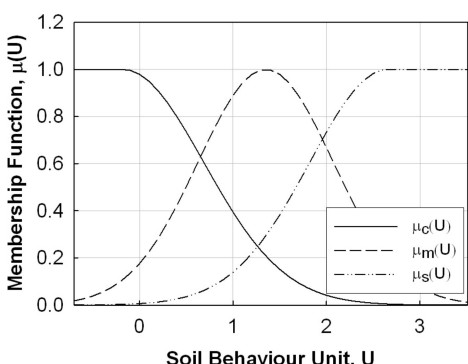

**Figure 3.** Fuzzy membership functions developed by Zhang and Tumay [7].

Since Pradhan [6] and Zhang and Tumay [7] developed fuzzy membership functions using a trial-and-error method according to the charts of Robertson et al. [3] and Douglas and Olsen [2], respectively, their method may result in very similar results to the soil classification result presented in the original chart. To overcome this, in this study, fuzzy memberships were determined using a neural network process.

## 3. Fuzzy Clustering and Neuro-Fuzzy Modeling

### 3.1. Database

The database for this study was built from 17 local sites of South Korea, as shown in Figure 4, along the coastal line, and six sites were used for verification of the model. The closed circle indicates the location where training data for training the fuzzy membership function were extracted, and the open square indicates the location where the verification data for verifying the completed fuzzy soil classification system were obtained. Table 1 summarizes the site location, number of PCPTs, and soil type classified by USCS. The measured values from piezocone penetration tests—$q_c$, $f_s$, $u_{bt}$—were averaged within the interval of 5 to 10 cm and picked up at the same depth where SPT and undisturbed samples were taken. The database contained 5173 data points in total. Table 2 shows the classification results of the database into six categories following USCS. A huge number of clayey soil (CH, CL) samples were, included while the number of silty soil and sandy soil samples was relatively small due to the difficulty in soil sampling with a thin wall tube sampler.

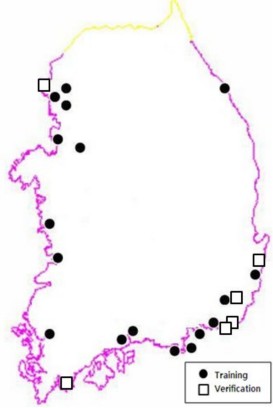

**Figure 4.** Seventeen local sites for this study.

**Table 1.** Included number of PCPTs and USCS classes for each site.

| Sites | | Nos. | Soil Type by USCS |
|---|---|---|---|
| Gyeonggi | Pyeongtaek | 2 | CL, SP, SW |
| | Siheung | 3 | CL, SM, SP |
| | Ilsan | 1 | SM, SP |
| | Incheon | 3 | CL, ML |
| Chungnam | Seocheon | 4 | CL, SM, SP |
| | Asan | 1 | CL, SM |
| Jeonnam | Yeongam | 1 | CH, SP, SW |
| | Gwangyang | 1 | CL, CH, ML, SP |
| Jeonbuk | Kunjang | 9 | CH, CL, ML |
| Gyeongnam | Yangsan | 4 | CL, SM |
| | Tongyeong | 2 | CH |
| | Hadong | 2 | CL, CH, MH, SP |
| | Ulsan | 2 | SM, SP-SC |
| | Yongwon | 4 | CH |
| | Cheonseong | 4 | CH |
| | Gaduk | 4 | CH |
| Kangwon | Naegok | 2 | CL, MH, SP, SW |

**Table 2.** Number of soil data points and corresponding PCPT data range.

| USCS | Nos. | $q_t$ (MPa) | $f_s$ (MPa) | $u_{bt}$ (MPa) |
|---|---|---|---|---|
| CH | 2746 | 0.108 to 1.250 | 0.0001 to 0.030 | 0.051 to 0.683 |
| CL | 1861 | 0.028 to 6.520 | 0.0001 to 0.052 | 0.002 to 1.088 |
| MH | 36 | 0.608 to 1.640 | 0.003 to 0.028 | 0.018 to 0.255 |
| ML | 284 | 0.436 to 6.818 | 0.004 to 0.096 | −0.092 to 0.562 |
| SM, SP-SC | 148 | 0.217 to 160.328 | 0.006 to 5.740 | −0.960 to 3.256 |
| SP, SW | 98 | 1.232 to 36.263 | 0.005 to 0.857 | −0.168 to 0.528 |

### 3.2. FCM (Fuzzy C-Means) Clustering Algorithm

Before developing the neuro-fuzzy model for PCPT-based soil classification, a grouping procedure was carried out to establish the unique structure between soil behavior type and PCPT input parameters, as well as to determine the appropriate number of soil types in the database and the input variables. Generally, some techniques exist to find structures in the database and divide them into small groups. An unsupervised learning strategy, clustering, can be used for that purpose, with the FCM (fuzzy C-means) algorithm being most widely used. This algorithm searches for fuzzy divisions $\bar{F} = \{\overline{F_1}, \overline{F_2}, \cdots, \overline{F_c}\}$ to minimize the function, as expressed in Equation (6), when a dataset composed of *n* items is divided into *c* clusters.

$$J_m(U, V : X) = \sum_{i=1}^{c} \sum_{k=1}^{n} (\mu_{ik})^m \|X_k - V_i\|^2, \tag{6}$$

where $V = \{V_1, V_2, \cdots, V_c\}$ is the set of *c* central vectors, and $\|X_k - V_i\|$ is the geometric distance between data $X_k$ and the center of the *i*th cluster. In addition, $\mu_{ik}$ is the grade of

cluster $\overline{F}_i$ including data $X_k$ and satisfies Equation (7) in the element of the fuzzy partition matrix $U = [\mu_{ij}]$ with the size of $(c \times n)$.

$$\mu_{ik} \in [0,1], \sum_{i=1}^{c} \mu_{ik} = 1 \tag{7}$$

The procedure of the FCM clustering algorithm is summarized below.

①    Assume partition number $c$ ($2 \leq c \leq n$) and fuzziness of partition $m$.

②    Select initial values of fuzzy partition matrix, $U(t)$. Random values are assumed for satisfying Equation (6).

③    Calculate the center of cluster $V$ using Equation (8).

$$V_i^{(t+1)} = \frac{\sum\limits_{i=1}^{c} \left(\mu_{ik}^{(t)}\right)^m X_k}{\sum\limits_{i=1}^{c} \mu_{ik}^{(t)}}, m > 1, i = 1, \cdots, c \tag{8}$$

④    Recompose fuzzy partition matrix using Equation (9).

$$\mu_{ik} = \frac{1}{\sum\limits_{j=1}^{c} \left(\frac{|X_k - V_i|^2}{|X_k - V_j|^2}\right)^{1/(m-1)}}, i = 1, \cdots, c, k = 1, \cdots, n \tag{9}$$

⑤    Complete the procedure if $|U(t+1) - U(t)| < \delta$. Otherwise, repeat phase ④. Here, $\delta$ is assumed to be $10^{-3}$.

⑥    Repeat phases ① to ⑤ and decide optimized partition number, as well as $c$ and $m$ values.

To determine the substructure of the compiled database and optimized input parameters, the success rate of any clusters and input parameters was evaluated after combining PCPT parameters, hydrostatic pressure, and total vertical stress, as shown in Table 3. First, the database was divided into 3–6 clusters to determine the optimized clusters of soil type according to combined parameters. Outputs were clay (CH, CL), silt (MH, ML), and sand (SM, SP, SP-SC, SW) for three clusters, clay (CH, CL), silt (MH, ML), sand with fine grained soil (SM, SP-SC), and relatively coarse sand (SP, SW) for four clusters, clay (CH, CL), silt with high liquid limit (MH), silt with low liquid limit (ML), sand with fine grained soil (SM, SP-SC), and coarse sand (SP, SW) for five clusters, and CH, CL, MH, ML, sand with fine grained soil (SM, SP-SC), and coarse sand (SP, SW) for six clusters. A total of 5 (input parameters) $\times$ 4 (clusters) were considered. The success rates of FCM clustering were evaluated as presented in Table 3. When all data points were concentrated in a specific cluster and appropriate clustering was not possible, is the success rate was regarded as "bad". According to the results, the maximum success rate was 74% when $q_t$, $R_f$, and $B_q$ were used as input parameters and three output clusters were selected. Furthermore, the m value representing the fuzziness of the partition was optimized, as given in Figure 5. Success rates were increased to $m = 4$ and seemed to converge after $m = 4$. Thus, $m = 4$ was adopted for this study.

**Table 3.** Success rate of FCM clustering for selected input parameters and specified clusters.

| Input Parameters | Success Rate (%) | | | |
|:---:|:---:|:---:|:---:|:---:|
| | Three Clusters | Four Clusters | Five Clusters | Six Clusters |
| $q_t$, $f_s$, $u_{bt}$ | 71 | 61 | 46 | Bad |
| $q_t$, $R_f$, $B_q$ | 74 | 60 | 48 | 42 |
| $q_t$, $f_s$, $\Delta u$ * | 70 | 60 | 69 | 42 |
| $q_t$, $f_s$, $u_{bt}$, $\sigma_{vo}$ | 71 | 53 | Bad | 52 |
| $q_t$, $f_s$, $u_{bt}$, $R_f$, $B_q$, $\sigma_{vo}$ | 70 | 58 | 48 | Bad |

* $\Delta u = u_{bt} - u_o$.

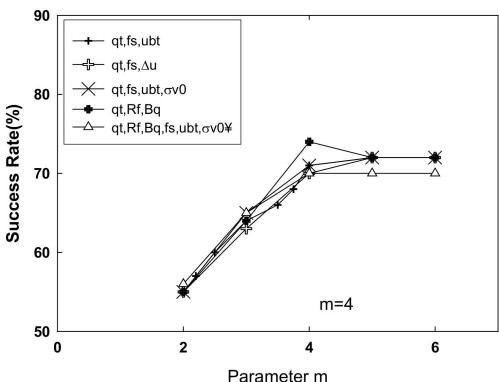

**Figure 5.** Change in success rate with respect to m value.

### 3.3. Neuro-Fuzzy Algorithm

After the FCM clustering described in Section 3.2, a neuro-fuzzy model was developed on the basis of the optimum number of clusters (three, i.e., clay, silt, and sand) and input variables ($q_t$, $R_f$, $B_q$). The fuzzy technique has the advantage of presenting data or events which cannot be numerically expressed, whereas the selection of the fuzzy membership function for inference is not always objective and precise. The decisions of fuzzy membership functions by Pradhan [6] and Zhang and Tumay [7] were dependent upon subjective trial-and-error methods or experimental methods. Thus, any revision or supplement considering local characteristics may not be easily considered even though it is necessary. However, the neuro-fuzzy method combining a neutral network and the fuzzy method has the merits of both techniques and is expected to overcome the previously mentioned defects. The neural network has the advantage of facing variations of data, but the input data are in numeric form. On the other hand, the fuzzy technique allows presenting numeric data using a membership function. Thus, the techniques are complementary. Moreover, the neuro-fuzzy technique can objectively decide membership functions and can be easily updated if needed through determining the optimized membership functions using the neural network algorithm. Figure 6 shows a schematic diagram of the ANFIS neuro-fuzzy model with two input parameters.

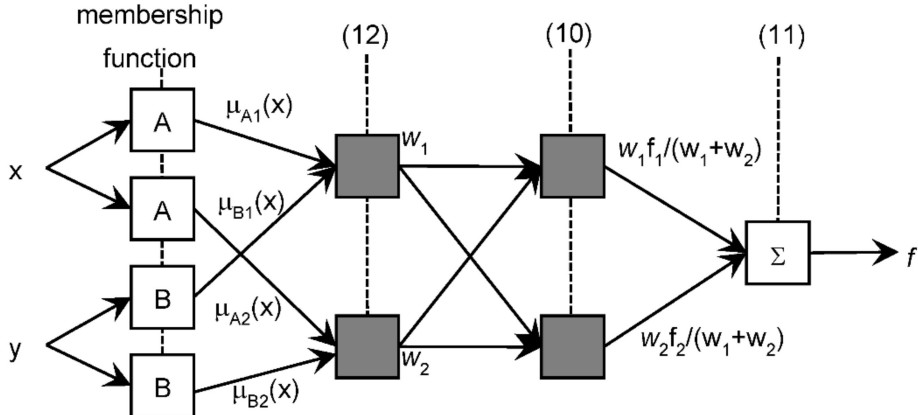

**Figure 6.** Procedure of ANFIS neuro-fuzzy model.

The general neuro-fuzzy model uses membership functions in the input and output phases. In this case, more time is required to complete calculation and convergence. Thus, the ANFIS (adaptive network-based fuzzy inference system) adopts a first-order function in the output instead of a membership function. The relevant procedure of the ANFIS model is summarized below.

①    The neuro-fuzzy output shown in Figure 6 is defined as a first-order function as shown in Equation (10) if two input parameters are assumed. Here, *p*, *q*, and *r* are constants to be decided after neural network training, while *x* and *y* are input parameters, which are PCPT indices.

$$f_1 = p_1 x + q_1 y + r_1,$$
$$f_2 = p_2 x + q_2 y + r_2,$$
$$f = f_1 + f_2$$

(10)

②    Total outputs in the system considering weighting factors $w_1$ and $w_2$ are given by Equation (11). Weighting factors are calculated using Equation (12) after evaluating each fuzzy membership function for given input parameters. Here, $\mu_{A1}$ corresponds to the finalized fuzzy membership function of *x*, while $\mu_{B1}$ corresponds to the finalized fuzzy membership function *y*.

$$f = \frac{w_1 f_1 + w_2 f_2}{w_1 + w_2}.$$

(11)

$$w_1 = \mu_{A1}\mu_{B1}, w_2 = \mu_{A2}\mu_{B2}.$$

(12)

③    The training procedure is completed after optimizing the parameters (*p*, *q*, and *r*) of the first-order function and of membership functions to minimize output error, *e*, defined by output $f_k$ and estimation $T_k$.

$$e = \frac{1}{2}\sum (f_k - T_k)^2.$$

(13)

Commercial soft computing package, Matlab was used to complete the training procedure when the least square error in Equation (13) was within the target error value, $\varepsilon = 0.01$ or when the maximum training loop attained 200. If any of the predetermined conditions were not satisfied, is the outcome was regarded as "bad". Figure 7 shows the commonly used candidate membership functions for the input process—i.e., triangular, Gaussian, bell-shapes, and sigmoidal (S-shaped) membership functions.

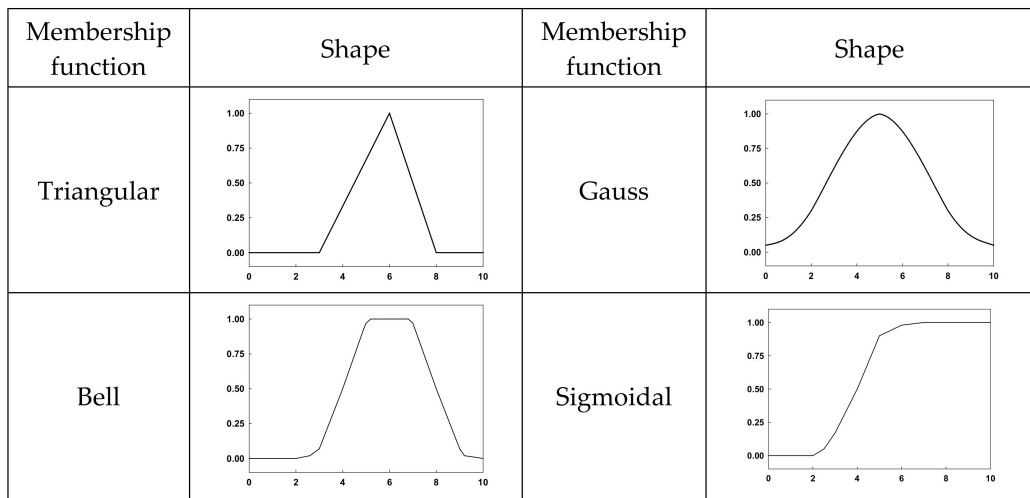

**Figure 7.** Shapes of fuzzy membership functions for input parameters.

Tables 4–7 show the various neuro-fuzzy analysis results to find the best combination of fuzzy membership functions for input variables. The success rate for each class, which is defined by the match with the soil type in the database, and the averaged success rates per each combination are presented. As shown in the tables, the success rate generally ranged from 70% to 79%. Among the results, the maximum success rate was 79.09% when the triangular membership function, Gaussian membership function, and sigmoidal membership

function were selected as the membership functions for $q_t$, $R_f$, and $B_q$ respectively. The optimized shapes of each membership function after training are shown in Figure 8.

**Table 4.** Success rate when using a triangular membership function for $q_t$.

| Selected Fuzzy Membership Functions | | | Success Rate (%) | | | |
|---|---|---|---|---|---|---|
| $q_t$ (MPa) | $R_f$ | $B_q$ | Clay | Silt | Sand | Average |
| Triangular | Triangular | Triangular | 99.88 | 60.31 | 75.61 | 78.60 |
| | Triangular | Gaussian | 99.43 | 57.19 | 72.76 | 76.46 |
| | Triangular | Bell | 99.58 | 55.31 | 76.83 | 77.24 |
| | Triangular | Sigmoidal | 99.63 | 57.81 | 75.61 | 77.68 |
| | Gaussian | Triangular | 99.85 | 55.06 | 80.49 | 78.80 |
| | Gaussian | Gaussian | 99.43 | 57.19 | 73.58 | 76.73 |
| | Gaussian | Bell | 99.63 | 54.38 | 77.24 | 77.08 |
| | Gaussian | Sigmoidal | 99.58 | 57.19 | 80.49 | 79.09 |
| | Bell | Triangular | 99.38 | 58.44 | 73.98 | 77.27 |
| | Bell | Gaussian | 99.48 | 55.94 | 75.61 | 77.01 |
| | Bell | Bell | 99.63 | 54.06 | 78.46 | 77.38 |
| | Bell | Sigmoidal | 99.60 | 55.94 | 80.89 | 78.81 |
| | Sigmoidal | Triangular | 99.43 | 57.5 | 73.98 | 76.97 |
| | Sigmoidal | Gaussian | 99.48 | 56.25 | 75.61 | 77.11 |
| | Sigmoidal | Bell | 99.58 | 53.13 | 77.24 | 76.65 |
| | Sigmoidal | Sigmoidal | 99.58 | 55.31 | 80.89 | 78.59 |

**Table 5.** Success rate when using a Gaussian membership function for $q_t$.

| Selected Fuzzy Membership Functions | | | Success Rate (%) | | | |
|---|---|---|---|---|---|---|
| $q_t$ (MPa) | $R_f$ | $B_q$ | Clay | Silt | Sand | Average |
| Gauss | Triangular | Triangular | 99.60 | 57.19 | 73.98 | 76.92 |
| | Triangular | Gaussian | 99.65 | 56.25 | 73.98 | 76.63 |
| | Triangular | Bell | 99.43 | 51.25 | 72.36 | 74.35 |
| | Triangular | Sigmoidal | 99.48 | 57.19 | 71.95 | 76.21 |
| | Gaussian | Triangular | Bad | Bad | Bad | Bad |
| | Gaussian | Gaussian | 99.65 | 57.50 | 76.42 | 77.86 |
| | Gaussian | Bell | 99.43 | 52.50 | 76.02 | 75.98 |
| | Gaussian | Sigmoidal | 99.48 | 55.63 | 78.05 | 77.72 |
| | Bell | Triangular | 99.13 | 54.69 | 72.36 | 75.39 |
| | Bell | Gaussian | 99.65 | 57.50 | 76.83 | 77.99 |
| | Bell | Bell | 99.43 | 52.50 | 74.80 | 75.58 |
| | Bell | Sigmoidal | 99.50 | 55.63 | 78.05 | 77.73 |
| | Sigmoidal | Triangular | 99.13 | 55.63 | 67.07 | 73.94 |
| | Sigmoidal | Gaussian | 99.58 | 55.63 | 76.02 | 77.08 |
| | Sigmoidal | Bell | 99.48 | 51.88 | 72.76 | 74.71 |
| | Sigmoidal | Sigmoidal | 99.50 | 55.31 | 76.42 | 77.08 |

**Table 6.** Success rate when using a bell-shaped membership function for $q_t$.

| Selected Fuzzy Membership Functions | | | Success Rate (%) | | | |
|---|---|---|---|---|---|---|
| $q_t$ (MPa) | $R_f$ | $B_q$ | Clay | Silt | Sand | Average |
| Bell | Triangular | Triangular | 98.93 | 53.13 | 63.41 | 71.82 |
| | Triangular | Gaussian | 99.60 | 55.00 | 72.76 | 75.79 |
| | Triangular | Bell | 99.50 | 53.13 | 73.58 | 75.40 |
| | Triangular | Sigmoidal | 99.43 | 57.19 | 69.51 | 75.38 |
| | Gaussian | Triangular | Bad | Bad | Bad | Bad |
| | Gaussian | Gaussian | 99.73 | 57.50 | 77.64 | 78.29 |
| | Gaussian | Bell | 99.58 | 55.31 | 78.46 | 77.78 |
| | Gaussian | Sigmoidal | 99.45 | 55.31 | 77.24 | 77.33 |
| | Bell | Triangular | 99.80 | 54.06 | 80.89 | 78.25 |
| | Bell | Gaussian | 99.60 | 55.94 | 75.61 | 77.05 |
| | Bell | Bell | 99.50 | 54.69 | 75.61 | 76.60 |
| | Bell | Sigmoidal | 99.48 | 54.38 | 76.42 | 76.76 |
| | Sigmoidal | Triangular | 98.98 | 55.63 | 65.04 | 73.22 |
| | Sigmoidal | Gaussian | 99.58 | 54.06 | 76.83 | 76.82 |
| | Sigmoidal | Bell | 99.45 | 50.31 | 71.14 | 73.63 |
| | Sigmoidal | Sigmoidal | 99.48 | 54.69 | 72.36 | 75.51 |

**Table 7.** Success rate when using a sigmoidal membership function for $q_t$.

| Selected Fuzzy Membership Functions | | | Success Rate (%) | | | |
|---|---|---|---|---|---|---|
| $q_t$ (MPa) | $F_R$ | $B_q$ | Clay | Silt | Sand | Average |
| Sigmoidal | Triangular | Triangular | 99.50 | 59.38 | 74.39 | 77.76 |
| | Triangular | Gaussian | 99.63 | 52.81 | 73.17 | 75.20 |
| | Triangular | Bell | 99.58 | 47.81 | 75.20 | 74.20 |
| | Triangular | Sigmoidal | 99.53 | 53.75 | 75.20 | 76.16 |
| | Gaussian | Triangular | 99.50 | 60.31 | 76.42 | 78.74 |
| | Gaussian | Gaussian | 99.60 | 53.44 | 74.80 | 75.95 |
| | Gaussian | Bell | 99.58 | 51.88 | 76.42 | 75.96 |
| | Gaussian | Sigmoidal | 99.55 | 52.81 | 76.83 | 76.40 |
| | Bell | Triangular | 99.53 | 59.06 | 78.46 | 79.02 |
| | Bell | Gaussian | 99.55 | 53.44 | 74.39 | 75.79 |
| | Bell | Bell | 99.58 | 52.50 | 74.80 | 75.63 |
| | Bell | Sigmoidal | 99.55 | 52.81 | 75.61 | 75.99 |
| | Sigmoidal | Triangular | 99.55 | 59.06 | 78.05 | 78.89 |
| | Sigmoidal | Gaussian | 99.60 | 52.50 | 77.24 | 76.45 |
| | Sigmoidal | Bell | 99.55 | 53.75 | 74.80 | 76.03 |
| | Sigmoidal | Sigmoidal | 99.55 | 52.81 | 73.98 | 75.45 |

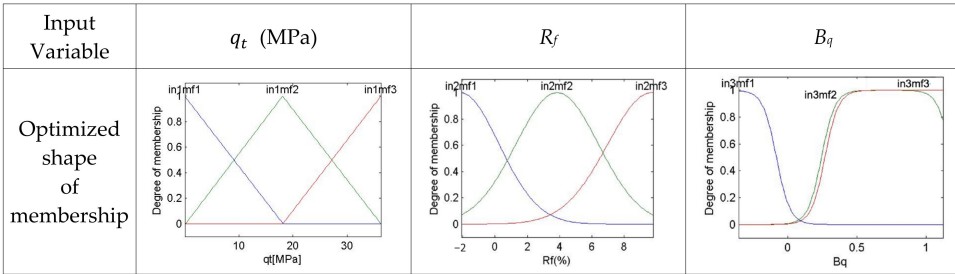

**Figure 8.** Optimized Shapes of fuzzy membership functions after training.

## 4. Verification of Suggested Neuro-Fuzzy Model

Verifications were performed with additional PCPT results which were not included in the training data, and the prediction results were compared to relevant boring logs. Piezocone tests for verification were carried out at Busan, Gyeongnam, and Jeonnam along the southern coast of Korea, Ulsan on the eastern coast of Korea, and Incheon on the western coast of Korea, as shown in Figure 4. Representative soil layers were three sites for clay, one for silt, and one for sand. For comparison, predictions from Pradhan's [6] and Zhang and Tumay's [7] methods using fuzzy theory and Robertson et al.'s chart [3] are also presented with the results of the newly suggested neuro-fuzzy model in this study. Soil classification results by Robertson et al. [3] were mainly related to soil behavior type, and their zones on the two charts were revised for the simplicity, i.e., clay for zone 3, silt for zones 4 and 5, sand for zones 8 and 9, and silt or sand for zones 6 and 7. Other zones not mentioned here are seldom found in South Korea (Kim et al., [20]). Indices for the results were as follows: 1 for clay, 2 for silt, and 3 for sand. In addition, zones 6 and 7 on Robertson's charts were marked as 2.5 and 0 for unclassified types.

### 4.1. Busan New Port Site

This site is located on the sea, and two PCPT penetrations were carried out. Soil layers were clay and a mixture of clay, sand, and gravel from the top of seabed. Thin silt lenses were found at the upper part of the deposit due to variations in seawater level. Laboratory test results from the undisturbed sample revealed the clay layer as highly compressible "CH". Water contents ranged from 51.5% to 75.3%, liquid limits ranged from 67.8% to 101.9%, and plastic limits ranged from 27.4% to 34.8%. Piezocone test results, boring logs, passing #200 sieve, water contents, and Atterberg limits are shown in Figures 9 and 10. From the results, Pradhan's method misclassified upper clay to some depth as silt, while Zhang and Tumay's method gave a better prediction of the narrow silt layer between clay at GL-16–17 m in Figure 9, but failed to predict the lower clay layer in Figure 10 as silt. Robertson's $q_t - B_q$ chart also provided satisfactory prediction when compared to boring logs, but had unclassified zones at the upper clay in both cases. On the other hand, the proposed neuro-fuzzy model from this study successfully classified the interbedded silt layer in Figure 9 and the clay layer in Figure 10. From the results, it was found that the proposed model provided more consistent classification with boring logs than others.

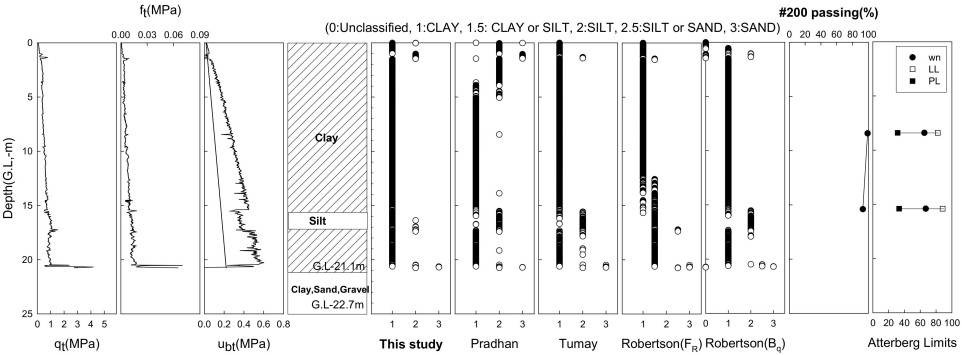

**Figure 9.** Verification results at Busan new port site-1.

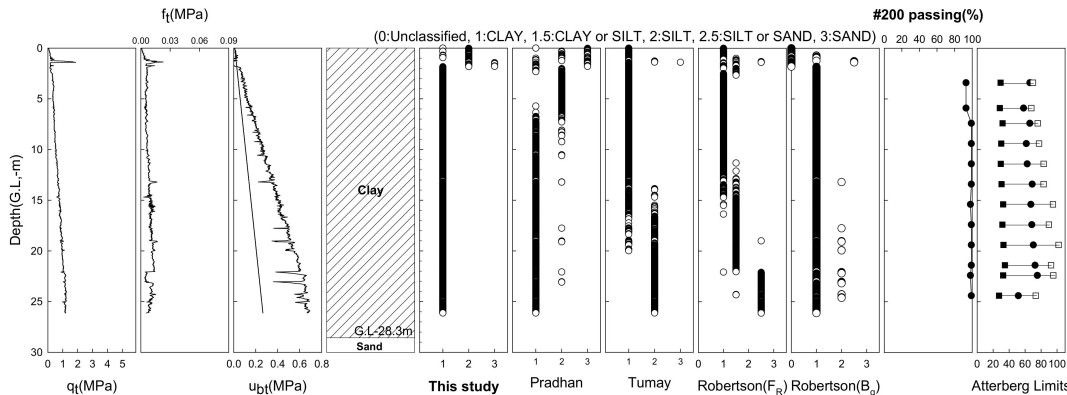

**Figure 10.** Verification results at Busan new port site-2.

### 4.2. Yangsan Site

According to the boring log, silty sand was distributed from surface to G.L-2.4 m and was layered by clay to G.L-11.7 m. A silt layer was interbedded in the clay layer at GL-5 m. A piezocone test was performed up to G.L-10.0 m. USCS results from undisturbed samples showed upper silty sand "CH" or "CL". Water contents ranged from 38.6% to 72.7%, liquid limits ranged from 36.0% to 68.6%, and plastic limits ranged from 17.0% to 26.9%. The classification results from every method are shown in Figure 11. Zhang and Tumay's method misclassified the silt layer and interbedded silt layer as clay due to the negative pore pressure measured at these layers. Kim et al. [20] reported this phenomenon whereby the precision of Zhang and Tumay's method is relatively low when negative pore pressure is measured because it does not incorporate the pore pressure index. Pradhan's method predicted the upper silty sand and mid-silty layer well, but failed to classify some upper clay right below the silty sand (circled zone) into silt. Robertson et al.'s estimation showed an unexpected result that it gave mainly unclassified points in the $B_q$ chart. However, the proposed method succeeded in classifying the upper silty sand and mid-silty layer, as well as the clay layer.

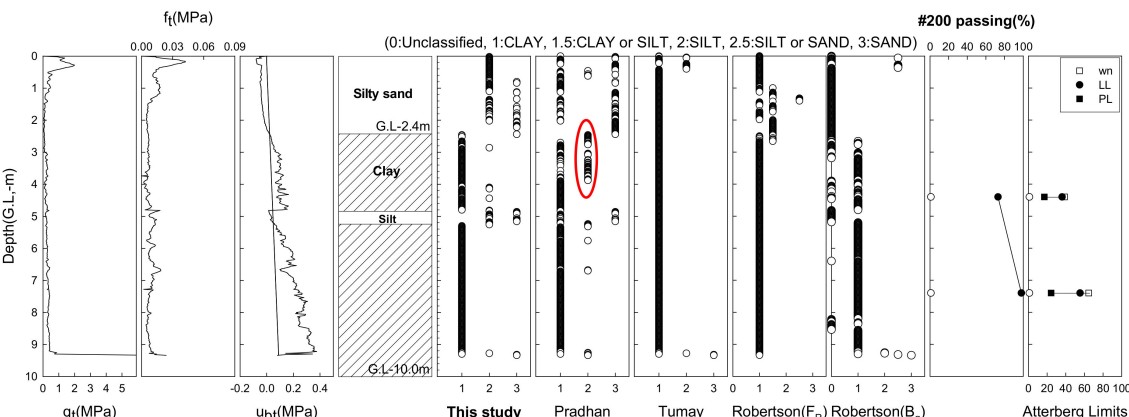

**Figure 11.** Verification results at Yangsan site.

### 4.3. Busan New Port Support Area Site

An 8 m thick intentional sand embankment above the clay layer was performed as a preloading for the purpose of accelerating consolidation as shown in Figure 12. Piezocone tests were performed to identify the bottom of the sand layer during the consolidation. Embanked sand was classified as "SM" from USCS. According to the results, Zhang and Tumay's method and the proposed neuro-fuzzy model yielded good agreement with the boring log. However, Robertson et al.'s classification from the $B_q$ chart gave unclassified points from 3 m to 7 m, as seen for the Yangsan site. Pradhan's method also revealed low

applicability when the pore pressure by PCPT was similar to hydrostatic pressure because its membership functions were derived from Robertson et al.'s charts.

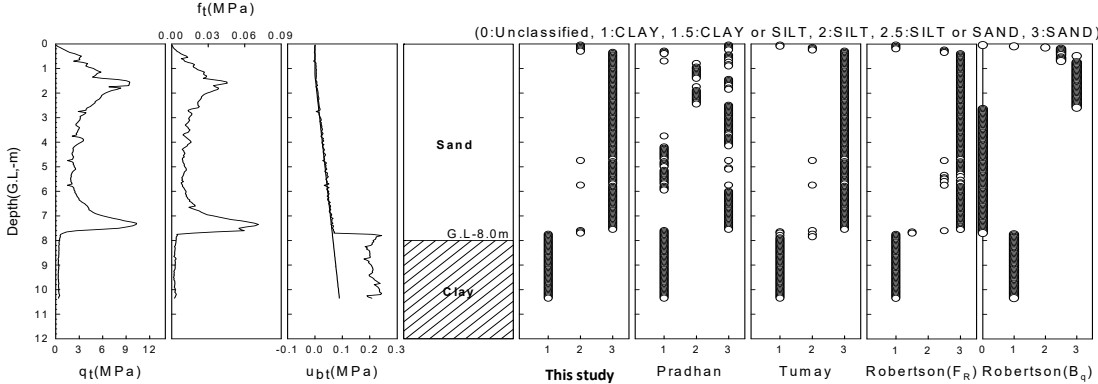

**Figure 12.** Verification results at Pusan new port support area site.

### 4.4. Jeonnam Dojang Port Site

This site is located on the southwestern coast of Korea on the sea. The top of the seabed was covered by silty sand up to G.L-1.8 m and layered by clay up to the depth of G.L-9.8 m as shown in Figure 13. A mixture of gravel and sand was distributed below clay. A casing tube was initially driven to G.L-0.5 m, and a piezocone test was carried out up to G.L-10 m. Soil classification results according to USCS from undisturbed samples in the clay layer were mainly "CH" or "CL". Water contents ranged from 29.59% to 65.27%, liquid limits ranged from 36.1% to 76.2%, and plastic limits ranged from 20.8% to 30.8%. Zhang and Tumay's method misclassified top silty sand mainly as clay because it did not consider negative pore pressure, as explained previously. Furthermore, Pradhan's method misclassified the upper part of "CH" or "CL" by USCS as silt. However, the neuro-fuzzy model classified clay and silty sand except at about 1 m thickness, but this may have occurred during the ground investigation considering the 1 m interval of SPT. The two charts proposed by Robertson et al. [3] failed to correctly detect silty sand. Moreover, the $R_f$ chart and $B_q$ chart gave different soil types in silty sand, which could confuse engineers.

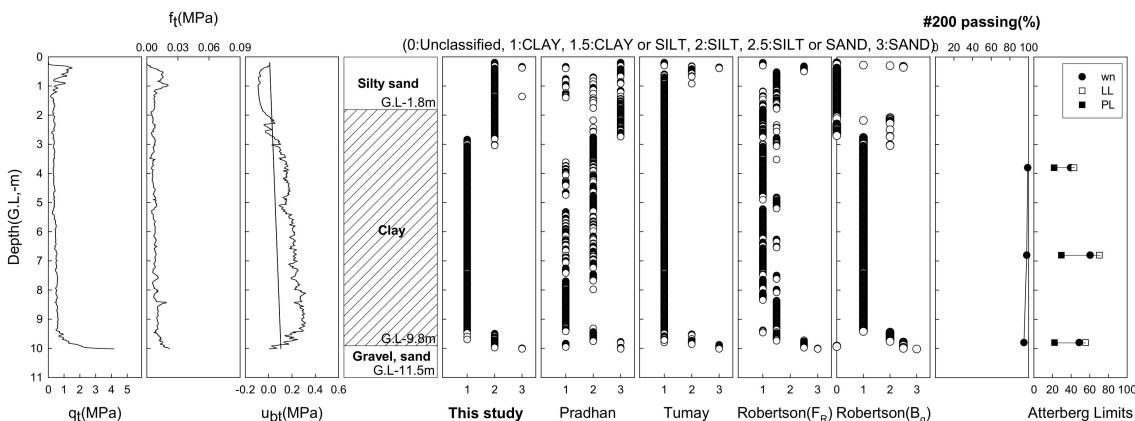

**Figure 13.** Verification results at Jeonnam Dojang new port site.

### 4.5. Incheon Trade Center Site

Incheon is located on the northwestern coast of Korea, where strong tidal action exists. The maximum tide difference is almost 9 m, and fine-grained soil is almost "ML". According to the boring log, fill reclaimed up to G.L-8.5 m and was layered by clayey silt up to G.L-19 m. Sandy silt and silty sand existed below the clayey silt. A piezocone test was performed to a depth of G.L-18.6 m through a fill into a casing tube. Soil classification

results according to USCS from undisturbed samples in the clayey silt layer were mainly "ML" due to tidal action, except for "CL" at around G.L-17 m to 18 m. Water contents ranged from 28.2% to 32.8%, liquid limits ranged from 34.2% to 38.9%, and plastic limits ranged from 21.9% to 29.5%. According to the results shown in Figure 14, Zhang and Tumay's method uniformly estimated all layers as silt and sand. The $B_q$ chart proposed by Robertson et al. [3] also classified clayey silt into silt or sand. Some misclassification was observed when considering the classification of USCS as "ML" or partially "CL". Pradhan's method estimated the mixture of clay, silt, and sand with depth and did not show any difference in detecting the thin clay layer at GL-17 m. The suggested neuro-fuzzy model also classified the clayey silt layer into mainly silt and succeeded in detecting the thin clay layer at around 17 m with "CL".

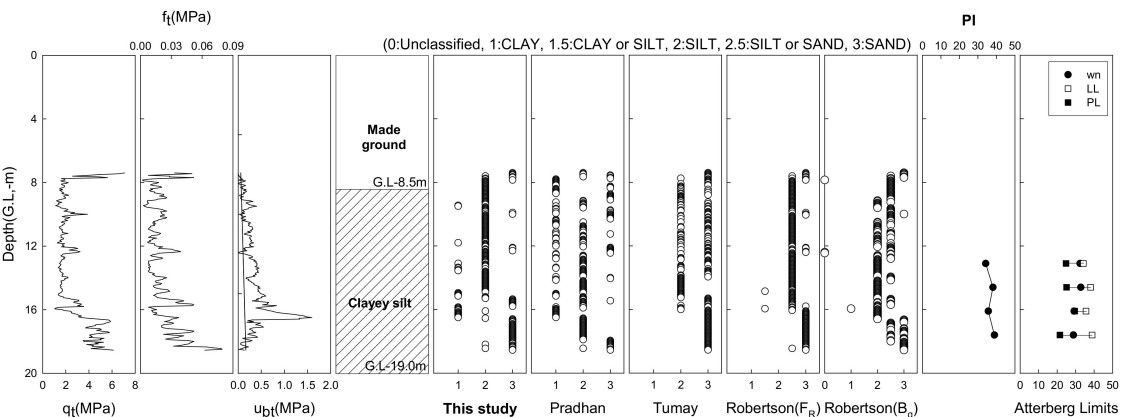

**Figure 14.** Verification results at Incheon trade center site.

### 4.6. Ulsan Southern Breakwater Site

This site is located on the southeastern coast of Korea, where the seawater depth is approximately 30 m. Clay was distributed to G.L-17.0 m and layered by sand or gravel. A casing tube was initially driven to G.L-3.7 m, and a piezocone test was carried out to G.L-16.7 m. The soil classification results of clay layer according to USCS from undisturbed samples were mainly "CH" through all depths, except for "CL" at G.L-16.0 to 16.8 m. Water contents of "CH" ranged from 70.1% to 90.8%, liquid limits ranged from 76.7% to 96.4%, and plastic limits ranged from 31.9% to 36.4%, whereas the water content of "CL" was about 38.1%, with a liquid limit of 41.2% and plastic limit of 21.9%. According to the classification results shown in Figure 15, all methods correctly classified the clay layer. However, Zhang and Tumay's method seemed to misclassify sand/gravel into silt since it did not appropriately reflect the variation of pore pressure, as mentioned above.

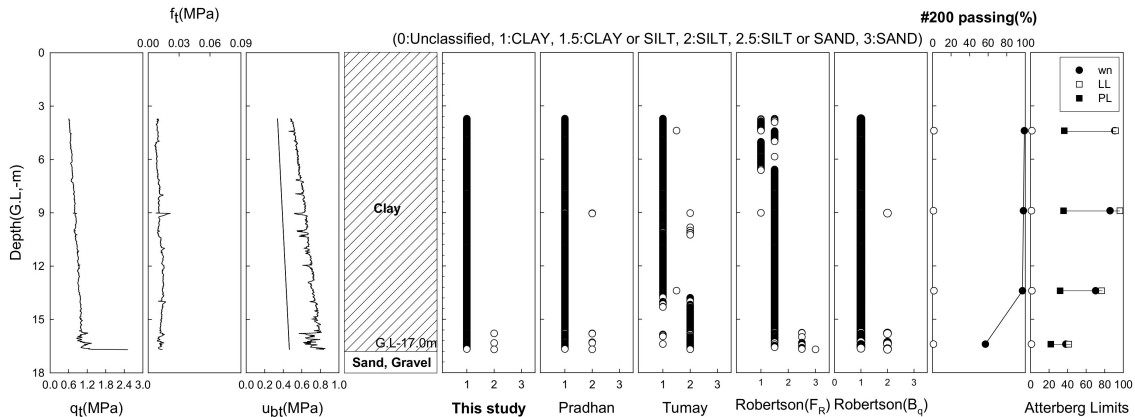

**Figure 15.** Verification results at Ulsan southern breakwater site.

## 5. Conclusions

A new soil classification system using FCM clustering and the neuro-fuzzy technique with piezocone test results was developed, and the main findings are summarized as follows:

(1) FCM clustering of the local database suggested that three input parameters of $q_t$, $R_{f,}$ and $B_q$ combined with three soil groups, i.e., clay, silt, and sand presented the highest success rate of 74.0%, and the m value representing the partition was optimized as 4.

(2) The neuro-fuzzy model was developed on the basis of the FCM clustering results with three input parameters $q_t$, $R_f$, and $B_q$ and three output classes, i.e., clay, silt, and sand. The training procedure was performed with a total of 5173 data points using various combinations of fuzzy membership functions. As a result, a maximum success rate of 79.09% was shown when the triangular membership function for $q_t$, Gaussian membership function for $R_f$, and sigmoidal membership function for $B_q$ were applied.

(3) Zhang and Tumay's method revealed low applicability when the penetration pore pressure by piezocone was negative or the same as the hydrostatic pressure since this method does not consider the pore pressure as an input parameter. The two charts presented by Robertson et al. sometimes failed to classify the upper reclaimed sand layer and interbedded sand or silt layer. Since Pradhan's method was adjusted to best match Robertson's diagram, both methods tended to yield essentially similar soil classification results. However, since Pradhan's method expressed a single soil classification using the overlapping fuzzy membership in Robertson's two diagrams and limited the maximum value of the fuzzy membership to 0.8, the soil classification result from Pradhan did not always match that of Robertson et al.

(4) The suggested neuro-fuzzy model matched well with boring logs and provided a better agreement with the classification in Korea. In addition, it has strong advantages in terms of revising or updating the model when the database is supplemented with new data.

**Author Contributions:** Conceptualization, Y.-S.K. and C.-H.K.; methodology, Y.-S.K. and C.-H.K.; validation, Y.-S.K. and J.-S.M.; investigation, C.-H.K.; writing—original draft preparation, Y.-S.K. and C.-H.K.; writing—review and editing, Y.-S.K. and J.-S.M.; supervision, Y.-S.K.; project administration, Y.-S.K.; funding acquisition, Y.-S.K. All authors have read and agreed to the published version of the manuscript.

**Funding:** This research was funded by the Ministry of Oceans and Fisheries, Korea, grant number 20180323.

**Institutional Review Board Statement:** Not applicable.

**Informed Consent Statement:** Not applicable.

**Data Availability Statement:** Not applicable.

**Conflicts of Interest:** The authors declare no conflict of interest.

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
