# Peer review of "Soil Classification from Piezocone Penetration Test Using Fuzzy Clustering and Neuro-Fuzzy Theory"

_applsci, doi:10.3390/app12084023_

Round 1
Reviewer 1 Report
This article presents a Combined AI model of Neural networks with Fuzzy theory for classifying the soil, using the data obtained from the cone penetration tests. This proposed new model developed is compared with the existing models and validated. New testing data from different locations are being tested. Very good and interesting work has been done.
There are a few minor corrections that are required in this article.
- All the literatures, models shown are dated back in the 1990s and early 2000s. There has been no literature referenced to the recent work done on this field, soil classification methods or so. Please include them
- The drawbacks of the existing models or methods are not described in detail. The state of the art methods available and why there is a need for this neuro-fuzzy model needs to be address in the intro section
- Line 68 - Typo "Prdhan" should be "Pradhan"
- Line 65 - Abbreviate FCM for its 1st use
- Section 2.2.1 and 2.2.2 are reusing content from the actual article, written in very large detail, rather it can be reduced and cited.
- Line 147 - Typo - missing "by"
- The reviewer does not notice the dataset size, like how many training and cross-validation dataset sizes were used.
Very good work on validation with the old Pradhans and Zhang Models.
Author Response
First of all, we would like to thank the reviewer for helpful advice and comments through the review.
Please find attached file.

Reviewer 2 Report
This paper presents a neuro-fuzzy model based on the FCM clustering results with three input parameters like qt, Rf and Bq and three output classes clay, silt and sand. Training procedure to some data was performed with the various combinations of fuzzy membership functions. Maximum success rate of 79.09% was shown when triangular membership function for qt, gauss mem-390 bership function for Rf and sigmoidal membership function for Bq were applied.
The proposed neuro-fuzzy model was validated with new data set which were not included in training data. For the sake comparison, present fuzzy models and Robertson et al.’s chart were applied. Zhang and Tumay’s method shows low applicability when penetration pore pressure by piezocone is negative or the same as hydrostatic pressure since its method does not consider pore pressure as input parameter. My conern here is the comparison is not very convincing given the different conditions. More analysis and comparison work should be discussed.
Two charts by Robertson et al. sometimes fail to classify upper reclaimed sand layer and interbedded sand or silt layer. Pradhan’s fuzzy model was also not effective in Korea where Robertson et al’s method fails. The underlying reasoning is not very straightforward to the reviewer. Some explanations should be provided.
Some of the figures are quite cluttered. They should be displayed separatly. In the current situation, it is not possible to see if the paper is printed on even A4 papers. I understand many readers would enjoy reading a paper version instead of a pdf file.
The introduction can also be improved by adding some novelty and highlighting the contribution against the difficulties.
Author Response

(The authors gave the same response as above.)

Round 2
Reviewer 2 Report
I have no further comment. The paper can be accepted in Appl. Sci.
Author Response
Dear reviewer
I appreciate your valuable comments and advice for my paper.
Best regards,
Youngsang Kim